# Feasibility of a new multifactorial fall prevention assessment and personalized intervention among older people recently discharged from the emergency department

**Bouke W. HEPKEMA**[1]\*, **Lydia KÖSTER**[1], **Edwin GELEIJN**[1], **Eva VAN DEN ENDE**[2], **Lara TAHIR**[2], **Johan OSTÉ**[3], **Bernard PRINS**[4], **Nathalie VAN DER VELDE**[5], **Hein VAN HOUT**[6], **Prabath W. B. NANAYAKKARA**[2]

**1** Department of Rehabilitation VUmc, Amsterdam UMC Location VUmc, Amsterdam, The Netherlands, **2** Department of Internal Medicine, Section General Internal Medicine Unit Acute Medicine, Amsterdam UMC Location VUmc, Amsterdam, The Netherlands, **3** Senior Policy Advisor Health Care, Amsterdam, The Netherlands, **4** General Practitioner Medical Center Gelderlandplein, Amsterdam, The Netherlands, **5** Department of Internal Medicine, Section of Geriatric Medicine, Amsterdam UMC Location AMC Academic Medical Center, Amsterdam, The Netherlands, **6** Department of General Practice and Elderly Care Medicine, Amsterdam Public Health Research Institute, Amsterdam UMC, Vrije Universiteit Amsterdam, Amsterdam, The Netherlands

\* b.hepkema@amsterdamumc.nl

## Abstract

### Background and importance

Falls among older people occur frequently and are a leading cause of Emergency department (ED) admissions, disability, death and rising health care costs. Multifactorial fall prevention programs that are aimed to target the population at risk have shown to effectively reduce the rate of falling and fall-related injuries in community-dwelling older people. However, the participation of and adherence to these programs in real life situation is generally low.

### Objective

To test the feasibility of a transitionally organized fall prevention assessment with accompanying personalized intervention initiated at the ED.

### Design, settings and participants

A process evaluation, of a non-randomized controlled pilot trial for implementing a transitionally organized multifactorial fall prevention intervention, was performed using the Reach, Effectiveness, Adoption, Implementation, Maintenance (RE-AIM) framework to gain insight into the barriers and facilitators of implementation. Older fallers (>70yrs) presenting at the ED were selected based on ZIP-code and after obtaining informed consent, data for the evaluation was collected through questionnaires and interviews. Furthermore, feedback was collected from the healthcare providers.

**Data Availability Statement:** All relevant data are within the paper and its Supporting information files.

**Funding:** We received the grand from ZonMW (grandnumber: 2006533, E 37500,-) and we received an additional grand form municipal health services Amsterdam (E15000,-).

**Competing interests:** The authors declare that they have no competing interests.

## Main results

The consent was obtained by 24 (70%) of the patients approached directly at the ED and 17 (26%) of the patients approached later by phone. Adherence to the protocol by the participants, clinical assessors and family practice were all more than 90%. After three months, nine (26%) of the participants had at least one recurrent fall: three (20%) patients in the intervention group and six (32%) in the control group.

## Conclusion

ED presentation due to a fall in older persons provides a window of opportunity for optimizing adherence to a multifactorial fall prevention program as willingness to participate was higher when the patients were approached at the ED during their stay. Implementing a transitionally organized multidisciplinary fall prevention program was successful with a high protocol adherence.

## The Netherlands trial register

NTR NL8142, November 8, 2019.

## Introduction

In recent years, 75% of the emergency departments (ED) in the Netherlands reported an increase in the (re)presentations of older patients, predominately due to falls [1]. Globally, falls are a leading cause of disability, death and rising health care costs resulting in a major public health problem [2]. In 2019, 109,000 patients aged 65 years and older visited the ED after a fall incident [3]. Of these accidents, 81% occurred in a private setting and accounted for more than 3000 deaths an increase of 40% in the last five years, resulting in falls being the number eight cause of death in the Netherlands [4]. A cohort study showed that, in people aged 70 years and older, fall-related injuries were the most frequent presenting complaint during weekday peak presentation times at a ED in the Netherlands. More notably, one in five returned to the ED within 30 days after being discharged, mostly due to a new fall incident [5].

Age is one of the key factors making people prone to falls, possibly as the combined result of intrinsic, pharmacologic, environmental, behavioral and activity related factors [6]. According to Dutch and international guideline advices for fall prevention, every older patient (65+) presenting at the ED after a fall should receive a multifactorial falls assessment [7]. However, care at the generally busy ED's is mostly disease-oriented instead of patient-oriented. Therefore, it often does not adequately address the complex care needs of the older patients [8]. Thus, it is of major importance to develop transitional acute care pathways that can deliver multifactorial fall assessment and prevention programs. For this, an earlier study showed that improvement of structured information exchange between care providers in the acute care chain and stimulation of a more generalist approach are needed. In addition, the use of an assessment tool was also recommended [9].

Multifactorial fall prevention programs have shown to effectively reduce the rate of falling and fall- related injuries in community-dwelling older people when aimed at the population at risk [10]. Community physiotherapists in the Netherlands play an important role in fall prevention by providing several evidence based physiotherapy-led fall prevention programs such as Otago (for individual patients) and 'In balans' (for groups). These single intervention

programs are effective in reducing falls [11]. Unfortunately, the participation in fall prevention interventions is very low (2% of at risk population) [12].

This project sought to test the feasibility of a transitionally organized fall prevention assessment with accompanying personalized intervention initiated at the emergency department (ED).

## Patients and methods

### Study design and setting

The feasibility study was a multicenter parallel-group non-randomized controlled trial. Participants were recruited in the ED of the Amsterdam University Medical Center (Amsterdam UMC, location VUmc and the BovenIJ Hospital (a teaching and nonteaching hospital respectively, both in Amsterdam, The Netherlands). The Medical Ethics Review Committee of the VUmc reviewed the research proposal, approved the project and decided that the Medical Research involving Human Subjects Act did not apply. The study is registries with trail number Trial NL8142 and written informed consent was obtained. The process evaluation was performed according to the Reach, Effectiveness, Adoption, Implementation, Maintenance (RE-AIM) framework [13]. The RE-AIM component "Maintenance" is not evaluated in this paper, as this warrants a longer follow up in a larger sample.

### Participants

Recruitment of participants took place between November 1, 2019 and January 31st, 2020. Patients of 70 years old and above visiting the ED with a low energetic fall related injury, what means there is no indication for in-hospital treatment, (semi) independent living and able to give informed consent were eligible for inclusion. Low-energy fall is defined as a result of falling from standing height or less, while high-energy trauma is defined as any other type of trauma (e.g. falling from height higher than standing height and motor vehicle accident). Exclusion criteria were: non- Dutch speaking, indication for in-hospital clinical treatment or planned discharge to a residential age care facility, participating in another fall prevention program, not able to sign informed consent and a high impact fall. If potential participants were missed at admission they were reached by phone the next day. Participants flow diagram is summarized in Fig 1.

### Intervention

**Intervention and control.** Patients living in the CHAGZ (cooperation for GP's in Amsterdam South) zip code area were allocated to the intervention group. Patients outside of the CHAGZ were allocated as controls. Participants in the control group received usual care. Patients in the intervention group received usual care plus a multifactorial fall-assessment, the interRAI Home Care (interRAI-HC), and thereafter a multifactorial fall-prevention program (e.g. physiotherapy, changes in medication). The fall-assessment was conducted at the patient's home, by healthcare professionals (Nine physiotherapists and occupational therapists) within a week after the ED visit. An overview of the time path and data collection in Fig 2.

**Fall-screening (interRAI-HC).** The interRAI-HC identifies amenable risks related to people's clinical conditions, functioning, lifestyle and behavior, social and physical environment. The interRAI-HC has a high inter rater reliability and good convergent validity [14] and is a comprehensive tool for healthcare professionals that captures all domains of vulnerability for persons in a complex care situation [15]. Twenty-two embedded CAPs (Clinical Action Points) help the assessor address areas in function and health that can be improved or

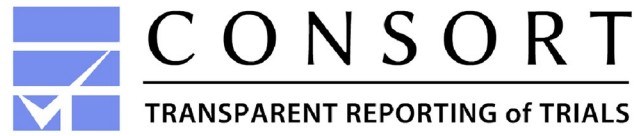

## CONSORT 2010 Flow Diagram

**Enrollment**

Assessed for eligibility (n=183 )

Excluded (n= 140 )
- Not meeting inclusion criteria (n=34 )
- Declined to participate (n= 63 )
- Other reasons (n= 43 )

Non- randomized (n=43)

**Allocation**

Allocated to intervention (n= 24)
- Received allocated intervention (n= 22 )
- Did not receive allocated intervention (secondary excluded (Bovenij hospital)) (n= 2)

Allocated to intervention (n= 19 )
- Received allocated intervention (n= 19)
- Did not receive allocated intervention (n= 0)

**Follow-Up**

Lost to follow-up (deceased, unable to be contacted, missing interRAI data) (n= 7)

Lost to follow-up (n= 0)

**Analysis**

Analysed (n=15)
- Excluded from analysis (n= 0)

Analysed (n= 19 )
- Excluded from analysis (n= 0)

**Fig 1. Flow diagram of elderly visiting ED after a low energetic fall.**

maintained. CAPS are based on systematic reviews of international literature, expert consensus, and analyses of large data holdings. They enable the assessor to develop a more responsive, individualized care plan. A triggered CAP can be considered as a 'red flag' marking an amenable health risk. The Falls CAP identifies people in whom a fall prevention program may be

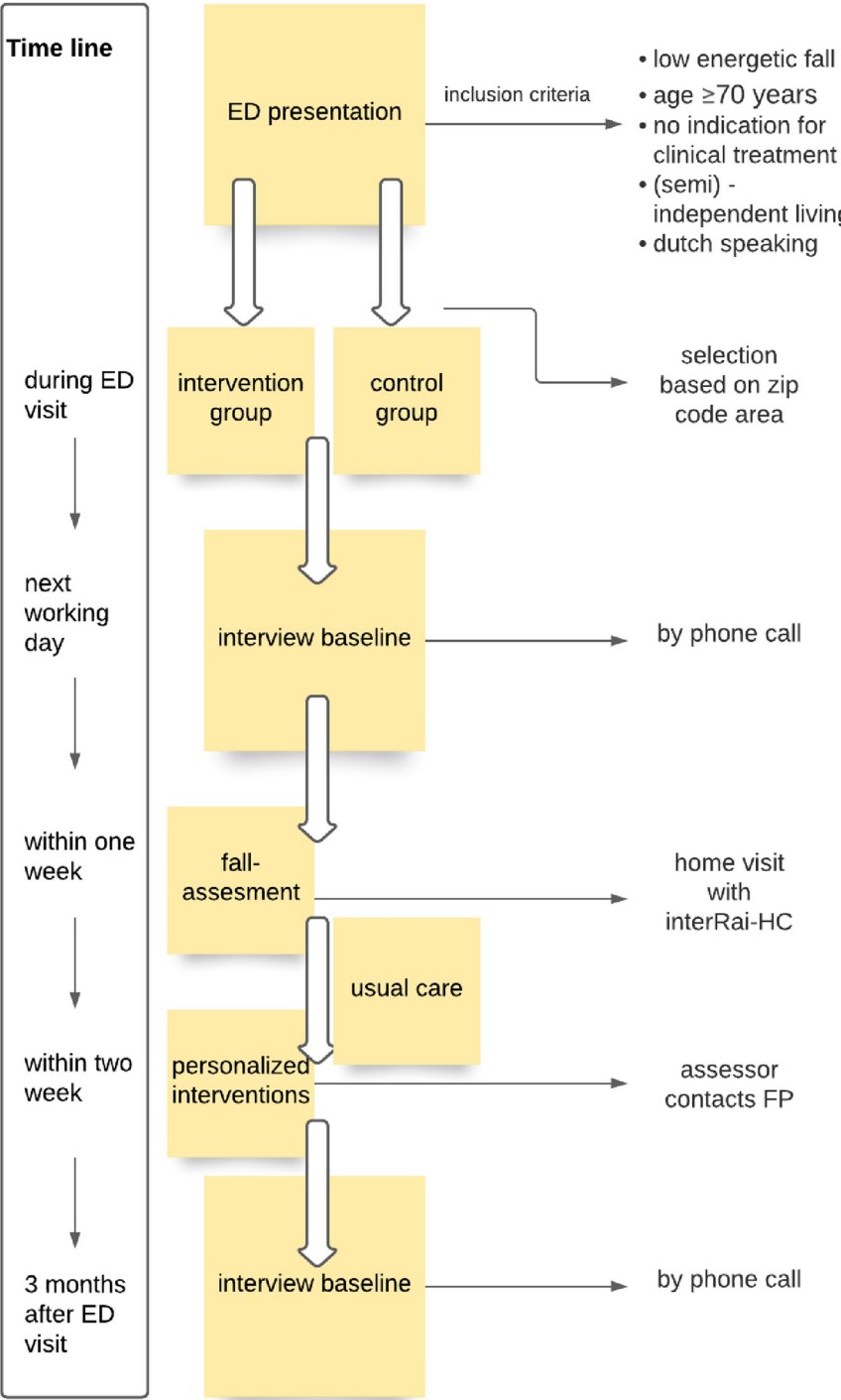

**Fig 2. Overview of time path and data collection.**

effective. The results of the assessment were interpreted and discussed with the participants reflecting a shared decision making process. The conclusions were forwarded to the family practice (FP) or the nurse practitioner for further action when indicated. The actions depended on the outcome of the assessment and the wishes of the participant and comprised,

among others, referral to a fall prevention assessment, change of medication or referral to an optician.

**Training of healthcare professionals.** The nine healthcare professionals were trained in three consecutive group sessions of two hours in performing and interpreting the inter-RAI-HC. The sessions were led by an experienced and specialized researcher. The assessors received a tablet to fill in the online assessments during their home visits.

## Data collection

Data were collected in a digital secured database (Castor EDC). The patients were interviewed by a researcher by phone one working day and three months after their ED visit. The interview consisted of questions about the fall (history), ED (re-)presentation, baseline characteristic (see Table 1) and the following questionnaires; I ADL [16] (Dutch questionnaire about activities in daily living) and short falls efficacy scale-international (short- FES) [17].

## Analysis

**Reach.** *Program reach* was evaluated by comparing the number of participants recruited into the study to the potential eligible patients presented the ED.

**Effectiveness.** The *number of recurrent falls* within three months after presentation.

**Adoption.** *Intervention participant adherence* was defined as the number of participants in the intervention group who received the InterRAI-HC assessment as planned within a week. *Communicated assessment results with family practice team* is defined as the number of assessments who were able to finish the whole assessment and communicated this information to the FP. *Clinical adherence FP or geriatric family nurse practitioner* was defined as the number of FP's or geriatric family nurse practitioner that were taking action after having received the assessment results.

**Implementation.** Implementation was evaluated by means of interviews and a focus group with healthcare professionals (a FP, geriatric nurse practitioner and two physiotherapists) and the research team for two hours. The participants in the intervention group were interviewed about their experiences three months after their initial ED visit. Also the FP or nurse practitioner was interviewed to evaluate initiated interventions and communication between health care professionals a month after the assessment. These interviews and the focus group provided insight in the feasibility of these transmural health care intervention. The opinions and experiences of healthcare professions (a FP, FP geriatric nurse practitioner and two physiotherapists) were collected.

## Results

A total of 41 participants were included, 22 in the intervention group and 19 in the control group. Participants in the intervention group had a mean (SD) age of 84 (9) years compared to a mean (SD) of 80 (6) years in the control group. In the intervention group four (17%) patients reported three or more falls in the last 12 months compared to two (11%) in the control group. In the intervention group 11 (48%) patients reported to be "never physically active" in the past 12 months compared to five (28%) in the control group. Many participants, n = 23 (52%) lived alone in the intervention and in the control n = 11 (61%) group. More than half of the intervention group n = 11 (58%) reported using four or more medication at time of fall compared to eight (44%) in the control group and 13 (57%) patients of the intervention group reported problems in cardiovascular pathology compared to eight (44%) in the control group. Baseline characteristics of participants are shown in Table 1.

**Table 1. Baseline characteristics.**

| Characteristics | All (N = 41) | Intervention (N = 22) | Control (N = 19) |
|---|---|---|---|
| Age (y), mean (SD) | 82 (8) | 84 (9) | 80 (6) |
| Age group, n (%) | | | |
| 70–80 | 18 (44%) | 8 (35%) | 10 (55%) |
| 80–90 | 16 (39%) | 9 (39%) | 7 (39%) |
| >90 | 7 (17%) | 6 (26%) | 1 (6%) |
| Female, n (%) | 24 (59%) | 14 (61%) | 10 (56%) |
| Alcohol use, n (%) | 25 (61%) | 12 (52%) | 13 (72%) |
| Tobacco use, n (%) | 1 (3%) | 0 (0%) | 1 (6%) |
| Lives alone, n (%) | 23 (56%) | 12 (52%) | 11 (61%) |
| First reported fall[1], n (%) | 17 (41%) | 8 (35%) | 9 (50%) |
| Number of falls[2], n (%) | | | |
| 0 fall | 24 (59%) | 10 (44%) | 14 (74%) |
| 1 fall | 7 (17%) | 7 (30%) | 0 (%) |
| 2 falls | 3 (7%) | 1 (4%) | 2 (11%) |
| 3 or more falls | 7 (17%) | 4 (17%) | 3 (16%) |
| Physically active[3] | | | |
| Daily | 9 (22%) | 5 (22%) | 4 (22%) |
| 3 times a week | 4 (10%) | 3 (13%) | 1 (6%) |
| Weekly | 10 (24%) | 3 (13%) | 7 (39%) |
| Monthly | 2 (5%) | 1 (4%) | 1 (6%) |
| Never | 16 (39%) | 11 (48%) | 5 (28%) |
| Mental status[4], n (%) | | | |
| Signs of depression | 9 (22%) | 4 (17%) | 5 (29%) |
| Less pleasure in activity's | 6 (15%) | 2 (9%) | 4 (23%) |
| Walking aid[5], n (%) | | | |
| Without walking aid, n (%) | 27 (66%) | 13 (56%) | 14 (77%) |
| With walking aid, n (%) | 13 (32%) | 10 (44%) | 3 (17%) |
| Wheelchair, n (%) | 1 (3%) | 0 (0%) | 1 (6%) |
| IADL total score[6], median [IQR] | 2 [0, 7] | 3 [0, 7.5] | 1 [0,4] |
| Short FES total score[7], median [IQR] | 8 [7, 9] | 8 [7,9] | 8 [7,10] |
| medical condition's reported by fall[8], n (%) | | | |
| arthritis | 1 (2%) | 0 (0%) | 1 (6%) |
| cardiac condition | 21 (51%) | 13 (57%) | 8 (44%) |
| respiratory condition | 1 (2%) | 1 (4%) | 0 (%) |
| diabetes | 5 (12%) | 3 (13%) | 2 (11%) |
| osteoporosis | 5 (12%) | 3 (13%) | 2 (11%) |
| stroke | 9 (22%) | 3 (13%) | 6 (33%) |
| other | 31 (76%) | 18 (78%) | 13 (72%) |
| Number of comorbidities[8], n (%) | | | |
| no comorbidities | 3 (7%) | 1 (5%) | 2 (10%) |
| 1–2 comorbidities | 13 (32%) | 6 (27%) | 7 (37%) |
| 3 comorbidities | 7 (17%) | 5 (23%) | 2 (10%) |
| 4 or more comorbidities | 18 (44%) | 10 (45%) | 8 (42%) |
| Number of prescription medication[8], n (%) | | | |
| no medication | 9 (20%) | 5 (22%) | 4 (21%) |
| 1–2 medications | 11 (27%) | 6 (26%) | 5 (26%) |
| 3 medications | 2 (5%) | 1 (4%) | 1 (5%) |

(*Continued*)

**Table 1.** (Continued)

| Characteristics | All (N = 41) | Intervention (N = 22) | Control (N = 19) |
|---|---|---|---|
| 4 or more medications | 19 (46%) | 10 (45%) | 9 (47%) |
| Medication reported by fall[8], n (%) | | | |
| sedative medication | 4 (10%) | 2 (9%) | 2 (11%) |
| antidepressant medication | 0 (%) | 0 (%) | 0 (%) |
| anti-epileptic medication | 2 (5%) | 1 (4%) | 1 (6%) |
| central analgesic medication | 8 (20%) | 4 (17%) | 4 (22%) |
| cardiac medication | 28 (68%) | 15 (65%) | 13 (72%) |
| long medication | 3 (7%) | 2 (9) | 1 (6%) |

[1] answer on the question during the interview 'is this your first fall?'.

[2] reported in last 12 months before this fall.

[3] physical activity status before fall, at least 20 min.

[4] mental status before fall, reported during interview day after ED visit.

[5] walking aid before fall reported during interview day after ED visit.

[6] IADL score range 0 to 21, low score means independent living is possible, high score independent living is not possible.

[7] Short FES total score range from 7 (no concern about falling) to 28 (severe. concern about falling).

[8] patient record research.

### Recruitment

The study started as a multicenter trial but the recruitment of participants at BovenIJ hospital was canceled due to organizational problems, there were two patients from BovenIJ hospital who had consented and completed the baseline assessment but both lost to follow up and therefore BovenIJ hospital was excluded from analyses. Older people presenting on the ED with a fall visited the ED most frequently between 3:00 PM and 00:00 AM.

### Reach

During the study period, 183 people age $\geq$ 70 years presented to the ED with a fall, of whom 34 (19%) were ineligible and 43 (23%) presented to the ED outside trial recruitment times and could not be reached by phone in the following days. Out of 106 eligible patients 63 (56%) did not want to participate in a fall prevention program or did not wish to be part of a research project. The consent was granted by 24 (70%) of the patients directly approached in the ED. Only 17 (26%) of the patients who were not invited during their ED visit but were approached by phone day later granted consent. The most common reason was that the fall was an accident and therefore the patients felt that they were not at risk for a new fall.

### Effectiveness

After three months, nine (26%) participants had at least one recurrent fall, three (20%) of them had been allocated to the intervention group, six (32%) to the control group. The sample size was too small to draw statistically sound conclusions.

### Adoption

*Intervention participant adherence*; from the 22 intervention participants, 21 had received the interRAI-HC assessment within one week (95%). *Clinical adherence interRAI assessor;* 9 out of 9 screeners were able to complete the whole screening process (100%). The assessment results were received properly by 17/18 nurse practitioners (94%), one nurse practitioner switched

job therefore the information was lost. *Communicated assessment results with family practice team;* one nurse practitioner received the assessment results but did not react (6%). Nevertheless 92% of the nurse practitioners followed the advices given by the assessor. Three education sessions of two hours were needed to obtain proper assessment results of the assessment group. Communication between ED and the assessors was fast by using a compliant and secure medical messaging platform (Siilo).

## Implementation

In four people (24%) there was no need to start any intervention based on the results of the interRAI-HC assessment which identified no fall risk factors. Three participants lived in the zip code area of the intervention group but had no FP in the area network and therefore the intervention could not be delivered. In 17 participants one or more interventions were started; Physiotherapy 5 (29%) participants, occupational therapy in 4 (24%), consultation visit by the nurse practitioner in 5 (29%), consultation visit FP 3 (18%), home care in 1 (6%). Table 2 shows an overview of all started interventions.

The interRAI-HC physical activity promotion indicator was triggered by 35% of the intervention participants, matching with the 29% of cases in which physiotherapy was started. The advice to prevent decline in cognitive function was triggered in 75% of the patients. However, in retrospect this conclusion was not taken up properly by assessors. Urinary incontinence was triggered to prevent decline by 10% and to facilitate improvement by 10%. A description of risks identified through Clinical Action Points in the interRAI-HC assessment instrument are shown in S1 Table in S1 Appendix [18]. The results of the CAPS from the interRAI-HC assessment from the intervention participants are shown in Fig 3 and in S2 Table in S1 Appendix.

## Identified barriers and facilitators by focus group discussion

Four main themes were identified in the analyses of the focus group. An overview of lessons learnt are shown in Table 3. The quotes and themes from the focus group discussions are shown in S3 Table in S1 Appendix.

## Communication between health professionals

The main barrier was the transmission of the results of the assessment from assessor to the FP or nurse practitioner, possibly the communication was not compelling enough. In future a compliant and secure medical messaging platform (Siilo) could be helpful, also a phone conversation a day after assessment was mentioned by a nurse practitioner.

## Motivation of the frail older person

Many patients underestimated the risk of recurrent falling: 'it was a one-off accident' or did not accept that they were frail. Nevertheless, in our project the least motivated people were

**Table 2. Interventions initiated after interRAI-HC assessment (N = 17).**

| Intervention | # | % |
|---|---|---|
| Physiotherapy | 5 | 29 |
| occupational therapy | 4 | 24 |
| Nurse practitioner visit | 5 | 29 |
| FP visit | 3 | 18 |
| Home care | 1 | 6 |

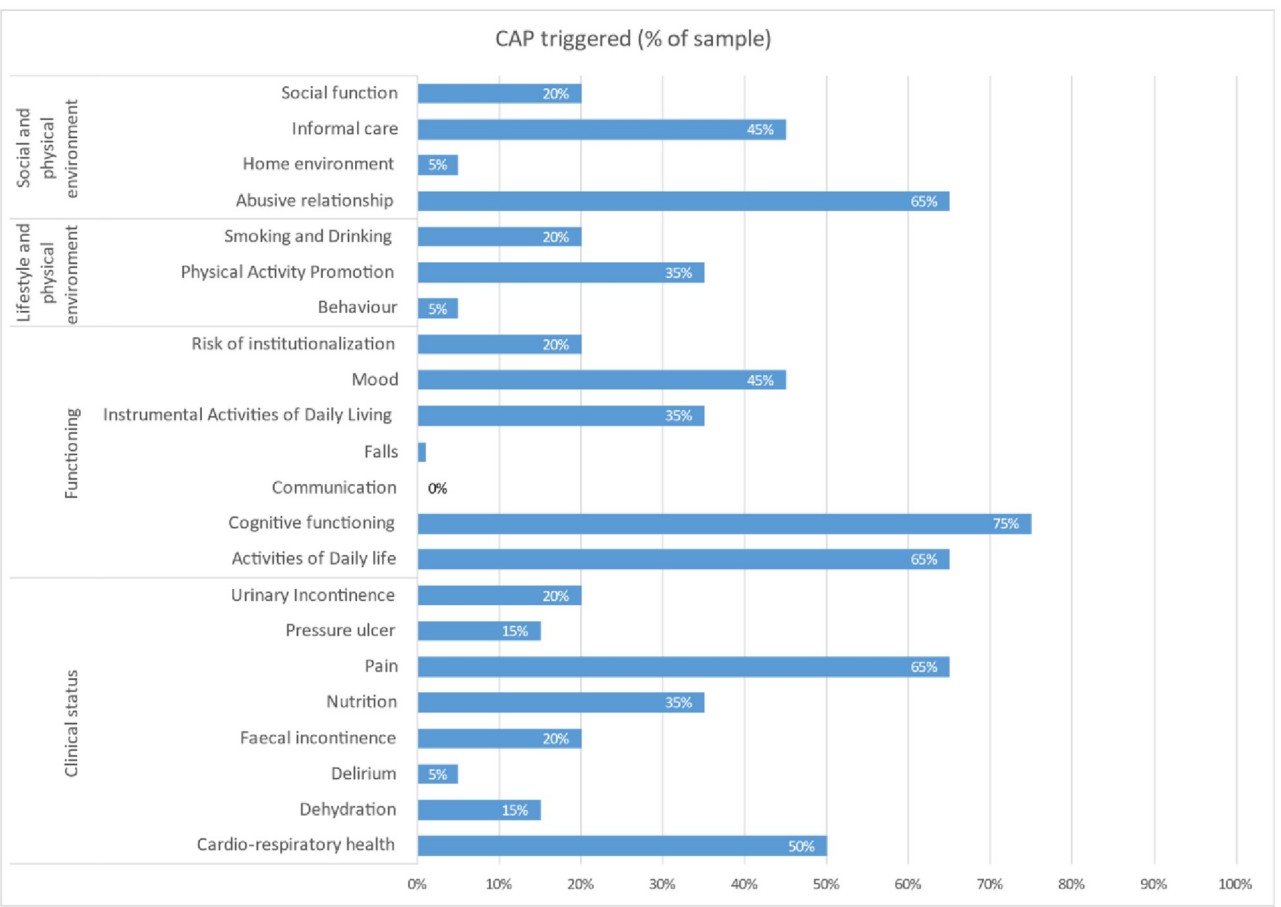

**Fig 3. An overview of the prevalence of triggered CAPs among older people who received an interRAI-HC assessment in primary care.**

**Table 3. Lessons learned.**

- **The ED is a promising location to identify older persons at risk of recurrent falling and invite patients to take part in a multifactorial fall prevention program.**
- **More refusers when approached by telephone compared to being approached at the ED.**
- **The refusers were mostly non-vulnerable patients or very vulnerable patients. Both categories have no indication for fall prevention.**
- **Older patients visit the ED after a fall mostly in the late afternoon and evening ($>$ 3 pm to approximately midnight).**
- **Adherence to the protocol by participants, assessors and family practice team is high.**
- **Communication between the health care professionals in the transmural acute care chain remains a challenge where using secure medical messaging platform could be helpful in optimizing communication.**
- **Three training sessions of two hours were needed to obtain proper assessment results of the assessment group.**
- **A lack of instructions to the nurse practitioners in terms of a health care coordinator limited the start of the interventions.**
- **Selection based on zip-code is possibly related with selection bias. Patients that present with a fall, that don´t live in the zip-code area of the ED tend to be less vulnerable and have less risk factors for recurrent falling. The term fall prevention deters people, it seems better to speak of a program that helps people to live longer independently at home.**

often the people who were possibly most at risk of falling, which makes it important to find means to motivate them to join at least a risk assessment. On the other hand the ED visit seems to provide a window of opportunity to motivate people to join fall prevention programs. During an ED visit, people are more aware of the danger of falling, which may make them more motivated to prevent this next time. Nurse practitioners mentioned that, involving a trusted and well known person to the patient could result in more motivation to join a program. Furthermore, motivational interviewing could be used as a tool in the process. The term fall prevention deters people, it seems better to speak of a program that helps people to live longer independently at home.

### InterRAI-HC assessment

The InterRAI-HC assessment is an extensive tool and the FP and nurse practitioners found it difficult to interpret the results. It requires a learning curve before it's clear how to properly conduct and interpret the assessment. Nevertheless when more experienced with interRAI-HC it a provides clear and fast insight. Contacting the nurse practitioner before performing the interRAI-HC to gauge whether patient is already in the picture at the family practice team, can give insight in the need of the assessment. Finally, for the assessment it appears enough to use the short interRAI-HC.

### Emergency department

The ED general perceived as a good location for identifying people for a fall prevention program because of the quick confrontation after the fall. Although, in the current care system ED nurses have a shortage of time for this. Possibly, in the future this must be incorporated in their regular tasks.

## Discussion

This project was set up to test the feasibility of a transitionally organized fall prevention assessment with accompanying personalized intervention initiated at the emergency department (ED). Our study showed that 70% of all eligible patients at the ED consented to participate, whereas only 26% of the patients who were contacted by phone call after ED presentation were willing to participate. The ED is therefore a promising window of opportunity to include patients for a multifactorial fall prevention program because of the acute setting and the sense of urgency. The adoption of the project was promising. There was a high adherence among participants, specialized physio- and occupational therapists and FP's, all more than 90%. A limiting factor in the transitional acute care chain appears to be the communication between assessor and the FP/nurse practitioner. Initially, some nurse practitioners were not prepared enough to take over the roll as the ´case manager´ or a "care coordinator" possibly hampering the start of the interventions.

Selection based on zip code resulted in unequal groups, patients from a zip code further away from the hospital (control group) are possibly daytrip people comparted to more vulnerable people who had a fall accident at home. This could explain the higher mortality in the intervention group (four people in the intervention group versus zero in the control group).

In the current study three subgroups of older fallers presenting at the ED were identified (data not shown). First, older persons with low risk for recurrent falling (defined as no treatable risk factors in the interRAI-HC). Second, the group with a high risk of recurrent falling but with terminal disease and indication for palliative care. Third, patients with high risk of recurrent falling who were deemed likely to benefit from the transitional multifactorial fall preventive intervention. These different groups need various approaches and preferably these

different groups should be identified during the screening process. A decision tree could then be used to yes/no offer the intervention and/or a simplified intervention such as exercise training and general falls preventive information (leaflet).

The major strength of this study is the successful roll out of a multidisciplinary network with more than 90% adherence to the protocol. Also the patient population is a good reflection of the intended population with a high risk of a recurrent fall. A limitation of the study was the possibly underreporting of falls due to recall bias. Assessing the number of falls by phone by older people appears to be suboptimal. Preferable to report the number of falls with a fall diary and a longer follow up, up to a year. In addition, for further roll-out assessor funding may be problematic for less affluent areas where patients are less additional insured. Possibly the geriatric nurse practitioner can do the assessments and reach a larger group of older patients.

These results have important implications for developing studies, for an overview see lessons learned in Table 3. Future research should be undertaken to start a multicenter randomized controlled trial with targeting in de ED, were the flaws of our study have been resolved. In our opinion each older faller presenting at the ED needs an fall risk assessment to address the individual risk of recurrent falling, which is in line with the new recommendations for specialized care (IGJ indicators) [7]. At the ED, identifying patients who would benefit from a fall prevention intervention can be difficult and a decision aid needs to be developed to identify which patients are likely to benefit from a multifactorial fall prevention programs. However, in the first instance at the ED it is just important to triage between too vulnerable or patients who would not benefit from a multifactorial fall prevention assessment. In our follow up project we plan to profile patients at the ED, with the clinical frailty scale [19].

Furthermore, communication between health care professionals in the transitional acute care chain can be difficult but necessary. Communication between healthcare professionals using secure medical messenger could be helpful in optimizing communication.

Besides this, the interRAI-HC is time consuming and because there is no physical test the objective information may be missing. In our opinion the short interRAI-HC in combination with a physical test such as the short physical performance battery may be preferable for the initial assessment [20]. Finally, the term "fall prevention" appears to have a deterrent effect. A recommendation would be to replace the words "fall prevention" and entice elderly patients to join by highlighting the following goal "living longer independently at home".

## Conclusion

ED presentation due to a fall provides a window of opportunity to identify and involve people with the risk of recurrent falling and motivate them to participate in a multifactorial fall assessment. The implementation of a transitionally organized multidisciplinary fall prevention network was successful. Although communication between health care professionals remained challenging. Digital communication platforms can help to improve interdisciplinary collaboration. The group of older persons visiting the ED after a fall are heterogeneous and not everyone is likely to benefit from a multifactorial prevention program. Thus identifying through screening who would and who would not be likely to benefit from such a multifactorial fall assessment remains important.

## Supporting information

**S1 Appendix.**
(DOCX)

**S1 File.**
(DOCX)

**S1 Checklist.**
(DOCX)

**S1 Data.**
(SAV)

## Acknowledgments

Focus group, the Patients, Family doctors and nurse practitioners of CHAGZ, specialized physio- and occupational therapists, ED nurses.

## Author Contributions

**Conceptualization:** Bouke W. HEPKEMA, Lydia KÖSTER, Edwin GELEIJN, Eva VAN DEN ENDE, Johan OSTÉ, Bernard PRINS, Nathalie VAN DER VELDE, Hein VAN HOUT, Prabath W. B. NANAYAKKARA.

**Data curation:** Bouke W. HEPKEMA, Lydia KÖSTER.

**Formal analysis:** Bouke W. HEPKEMA.

**Funding acquisition:** Bouke W. HEPKEMA.

**Methodology:** Bouke W. HEPKEMA, Hein VAN HOUT.

**Project administration:** Lara TAHIR.

**Supervision:** Nathalie VAN DER VELDE, Hein VAN HOUT, Prabath W. B. NANAYAKKARA.

**Validation:** Bouke W. HEPKEMA, Nathalie VAN DER VELDE.

**Writing – original draft:** Bouke W. HEPKEMA.

**Writing – review & editing:** Edwin GELEIJN, Lara TAHIR, Nathalie VAN DER VELDE, Hein VAN HOUT, Prabath W. B. NANAYAKKARA.

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
