## [Decision Letter · Decision Letter 0]

10 Feb 2022

PONE-D-21-36533Feasibility of a new multifactorial fall prevention assessment and personalized intervention among older people recently discharged from the emergency department.PLOS ONE

Dear Dr. Hepkema,

Thank you for submitting your manuscript to PLOS ONE. After careful consideration, we feel that it has merit but does not fully meet PLOS ONE’s publication criteria as it currently stands. Therefore, we invite you to submit a revised version of the manuscript that addresses the points raised during the review process.

We look forward to receiving your revised manuscript.

Kind regards,

Walid Kamal Abdelbasset, Ph.D.

Academic Editor

PLOS ONE

Journal Requirements:

2. We note that you have selected “Clinical Trial” as your article type. PLOS ONE requires that all clinical trials are registered in an appropriate registry (the WHO list of approved registries is at      https://www.who.int/clinical-trials-registry-platform/network/primary-registries" https://www.who.int/clinical-trials-registry-platform/network/primary-registries and more information on trial registration is at http://www.icmje.org/about-icmje/faqs/clinical-trials-registration/). 

Please state the name of the registry and the registration number (e.g. ISRCTN or ClinicalTrials.gov) in the submission data and on the title page of your manuscript.

a) Please provide the complete date range for participant recruitment and follow-up in the methods section of your manuscript.

b) If you have not yet registered your trial in an appropriate registry, we now require you to do so and will need confirmation of the trial registry number before we can pass your paper to the next stage of review. Please include in the Methods section of your paper your reasons for not registering this study before enrolment of participants started. Please confirm that all related trials are registered by stating: “The authors confirm that all ongoing and related trials for this drug/intervention are registered”.

Please see http://journals.plos.org/plosone/s/submission-guidelines#loc-clinical-trials for our policies on clinical trials.

3. We note that you refer to a Clinical Trial in your manuscript. Please include the name of the registry and the registration number (e.g. ISRCTN or ClinicalTrials.gov) in your manuscript. If the results have been previously reported, please provide a reference to the publication. 

4. Registration done retrospectively 

Thank you for submitting your clinical trial to PLOS ONE and for providing the name of the registry and the registration number. The information in the registry entry suggests that your trial was registered after patient recruitment began. PLOS ONE strongly encourages authors to register all trials before recruiting the first participant in a study.

1) your reasons for your delay in registering this study (after enrolment of participants started);

2) confirmation that all related trials are registered by stating: “The authors confirm that all ongoing and related trials for this drug/intervention are registered”.

7. We note that you have indicated that data from this study are available upon request. PLOS only allows data to be available upon request if there are legal or ethical restrictions on sharing data publicly. For information on unacceptable data access restrictions, please see http://journals.plos.org/plosone/s/data-availability#loc-unacceptable-data-access-restrictions. 

9. Please include a separate caption for each figure in your manuscript.

Reviewers' comments:

Reviewer's Responses to Questions

**Comments to the Author**

1. Is the manuscript technically sound, and do the data support the conclusions?

Reviewer #1: Yes

Reviewer #2: Partly

2. Has the statistical analysis been performed appropriately and rigorously? 

Reviewer #1: Yes

Reviewer #2: Yes

3. Have the authors made all data underlying the findings in their manuscript fully available?

Reviewer #1: Yes

Reviewer #2: Yes

4. Is the manuscript presented in an intelligible fashion and written in standard English?

Reviewer #1: Yes

Reviewer #2: Yes

5. Review Comments to the Author

Reviewer #1: This is a pilot study to evaluate feasibility. Statistics are descriptive and no formal hypothesis test can be performed. I just have minor concerns:

Table 1.

% for number of falls are not correct. Sum <100% and this variable should not have missing values.

% for number of comorbidities do not seem correct. Both treat and control summed more than 100%.

% for number of medications, more than 100%.

IADL use dot for decimal number.

Reviewer #2: The authors used a non-randomized controlled pilot trial of older fallers (>70yrs) presenting to test the feasibility of a transitionally organized fall prevention assessment with accompanying personalized intervention initiated at the ED, and found that ED presentation due to a fall in older persons provided a window of opportunity for optimizing adherence to a multifactorial fall prevention program and Implementing a transitionally organized multidisciplinary fall prevention program was successful with a high protocol adherence. The study provides data support for preventing falls in the elderly, and has certain clinical significance. However, the following concerns should be addressed:

1. There are many elderly fall patients in the emergency department, but the cases collected in the two groups in this paper are both small. I see that the author only collected the cases for 3 months, so I want to know why not collect more cases for a longer time? The results might be more convincing.

2. In line 105, could you explain a low energetic fall related injury in detail? How did you evaluate that?

Thank you!

6. PLOS authors have the option to publish the peer review history of their article (what does this mean?). If published, this will include your full peer review and any attached files.

Reviewer #1: No

Reviewer #2: No

---

## [Author Response · Author response to Decision Letter 0]

20 Apr 2022

Thank you for your comments on our article entitled “Feasibility of a new multifactorial fall prevention assessment and personalized intervention among older people recently discharged from the emergency department”. 

We confirm that the required revisions have been performed. Below a point by point response to reviewers comments: 

1) We note that you have selected “Clinical Trial” as your article type. PLOS ONE requires that all clinical trials are registered in an appropriate registry

Registered on November 8, 2019, registration number; NTR NL8142 

2) Registration done retrospectively 

The final conformation at the website was indeed performed a week after the start of the study. However, we were busy with the registration long before we started the study. 

3) We note that the grant information you provided in the ‘Funding Information’ and ‘Financial Disclosure’ sections do not match. 

We received the grand from ZonMW (grandnumber: 2006533, E 37500,- ) and we received an additional grand from municipal health services Amsterdam (E15000,- ). 

4) We note that you have indicated that data from this study are available upon request. PLOS only allows data to be available upon request if there are legal or ethical restrictions on sharing data publicly. 

We will submit the available data. 

5) Review Comments to the Author; Reviewer #1: This is a pilot study to evaluate feasibility. Statistics are descriptive and no formal hypothesis test can be performed. I just have minor concerns:

Table 1.

% for number of falls are not correct. Sum <100% and this variable should not have missing values.

% for number of comorbidities do not seem correct. Both treat and control summed more than 100%.

% for number of medications, more than 100%.

IADL use dot for decimal number.

The incorrect sum calculation in table one have been solved (see manuscript). 

6) There are many elderly fall patients in the emergency department, but the cases collected in the two groups in this paper are both small. I see that the author only collected the cases for 3 months, so I want to know why not collect more cases for a longer time? The results might be more convincing.

The reason to only collect cases for three months is, because in this qualitative study, after three months we reached data saturation. 

7) In line 105, could you explain a low energetic fall related injury in detail? How did you evaluate that?

Low-energy fall is defined as a result of falling from standing height or less, while high-energy trauma is defined as any other type of trauma (e.g. falling from height higher than standing height and motor vehicle accident). With the changing patient mix presenting at the emergency departments we see an increase influx of elderly with a low energetic fall related injury contrary to high energetic trauma related to road traffic accidents and violence. Care chain for high energetic trauma is well developed in the Netherlands while care-pathways for increasing low-energy trauma is not implemented well yet.

---

## [Decision Letter · Decision Letter 1]

6 May 2022

Feasibility of a new multifactorial fall prevention assessment and personalized intervention among older people recently discharged from the emergency department.

PONE-D-21-36533R1

Dear Dr. Hepkema,

We’re pleased to inform you that your manuscript has been judged scientifically suitable for publication and will be formally accepted for publication once it meets all outstanding technical requirements.

Kind regards,

Walid Kamal Abdelbasset, Ph.D.

Academic Editor

PLOS ONE

Additional Editor Comments (optional):

Reviewers' comments:

Reviewer's Responses to Questions

**Comments to the Author**

1. If the authors have adequately addressed your comments raised in a previous round of review and you feel that this manuscript is now acceptable for publication, you may indicate that here to bypass the “Comments to the Author” section, enter your conflict of interest statement in the “Confidential to Editor” section, and submit your "Accept" recommendation.

Reviewer #1: All comments have been addressed

Reviewer #2: All comments have been addressed

2. Is the manuscript technically sound, and do the data support the conclusions?

Reviewer #1: (No Response)

Reviewer #2: Yes

3. Has the statistical analysis been performed appropriately and rigorously? 

Reviewer #1: (No Response)

Reviewer #2: Yes

4. Have the authors made all data underlying the findings in their manuscript fully available?

Reviewer #1: (No Response)

Reviewer #2: Yes

5. Is the manuscript presented in an intelligible fashion and written in standard English?

Reviewer #1: (No Response)

Reviewer #2: Yes

6. Review Comments to the Author

Reviewer #1: (No Response)

Reviewer #2: The author has done a good job of answering all our questions. I have no more questions. Thank you very much!!

7. PLOS authors have the option to publish the peer review history of their article (what does this mean?). If published, this will include your full peer review and any attached files.

Reviewer #1: No

Reviewer #2: No

---

## [Editor Report · Acceptance letter]

31 May 2022

PONE-D-21-36533R1 

Feasibility of a new multifactorial fall prevention assessment and personalized intervention among older people recently discharged from the emergency department. 

Dear Dr. Hepkema:

I'm pleased to inform you that your manuscript has been deemed suitable for publication in PLOS ONE. Congratulations! Your manuscript is now with our production department. 

Kind regards, 

on behalf of

Dr. Walid Kamal Abdelbasset 

Academic Editor

PLOS ONE